# Impact of SARS-CoV-2 Infection on Patients with Cancer: Retrospective and Transversal Studies in Spanish Population

**DOI:** 10.3390/cancers12123513

**Published:** 2020-11-25

**Authors:** Javier Garde-Noguera, M. Leonor Fernández-Murga, Vicent Giner-Bosch, Victoria Dominguez-Márquez, José García Sánchez, Juan José Soler-Cataluña, Franscica López Chuliá, Beatriz Honrubia, Nuria Piera, Antonio Llombart-Cussac

**Affiliations:** 1Medical Oncology Department, Hospital Arnau de Vilanova, Fundación para el Fomento de la Investigación Sanitaria i Biomédica de la Comunidad Valenciana (FISABIO), 46020 Valencia, Spain; garcia_jossanc@gva.es (J.G.S.); honrubia_bea@gva.es (B.H.); piera_nur@gva.es (N.P.); antonio.llombart@medsir.org (A.L.-C.); 2Centre For Quality and Change Management, Universitat Politècnica de València, 46022 Valencia, Spain; vigibos@eio.upv.es; 3Microbiology Department, Hospital Arnau de Vilanova, Fundación para el Fomento de la Investigación Sanitaria i Biomédica de la Comunidad Valenciana (FISABIO), 46020 Valencia, Spain; dominguez_vicmar@gva.es; 4Pneumology Department, Hospital Arnau de Vilanova, Fundación para el Fomento de la Investigación Sanitaria i Biomédica de la Comunidad Valenciana (FISABIO), 46020 Valencia, Spain; soler_juacat@gva.es; 5Hematology Department, Hospital Arnau de Vilanova, Fundación para el Fomento de la Investigación Sanitaria i Biomédica de la Comunidad Valenciana (FISABIO), Medicine Department, Cardenal Herrera University, 46020 Valencia, Spain; francisca.lopez@uchceu.es; 6Universidad Catolica de Valencia “San Vicente Martir”, 46020 Valencia, Spain; 7Medica Scientia Innovation Research (MedSIR), 08018 Barcelona, Spain; 8Medica Scientia Innovation Research (MedSIR), Ridgewood, NJ 07450, USA

**Keywords:** SARS-CoV-2, COVID-19, cancer, chemotherapy, immunotherapy, targeted therapy, seroprevalence

## Abstract

**Simple Summary:**

Severe Acute Respiratory Syndrome Coronavirus 2 (SARS-CoV-2) pandemic has impacted the world, generating a global health emergency. There is concern about whether cancer patients represent a especially vulnerable to population to this disease, and about the potential influence of different antineoplastic treatments on the risk of contracting the infection and its evolution. With this work, we intend to evaluate the impact of the first wave of the pandemic in our population of cancer patients, analyzing the characteristics and evolution of those patients admitted at our hospital with confirmed diagnosed of SARS-CoV-2 infection, as well as the presence of symptoms and acquired seroprevalence among patients under treatment with different antineoplastic drugs. This knowledge could help to optimize cancer patient management during this period of time, providing information on the risk and outcome of the infection for our patients, and the safety of cancer treatments.

**Abstract:**

Background: Studies of patients with cancer affected by coronavirus disease 2019 (COVID-19) are needed to assess the impact of the disease in this sensitive population, and the influence of different cancer treatments on the COVID-19 infection and seroconversion. Material and Methods: We performed a retrospective analysis of all patients hospitalized with RT-PCR positive for COVID-19 in our region to assess the prevalence of cancer patients and describe their characteristics and evolution (Cohort 1). Concurrently, a transversal study was carried out in patients on active systemic cancer treatment for symptomatology and seroprevalence (IgG/IgM by ELISA-method) against Severe Acute Respiratory Syndrome Coronavirus 2 (SARS-CoV-2) (Cohort 2). Results: A total of 215 patients (Cohort 1) were admitted to hospital with a confirmed COVID-19 infection between February 28 and April 30, 2020, and 17 died (7.9%). A medical record of cancer was noted in 43 cases (20%), 6 of them required Intensive care unit ICU attention (14%), and 7 died (16%). There were thirty-six patients (83%) who tested IgG/IgM positive for SARS-CoV-2. Patients on immunosuppressive therapies presented a lower ratio of seroconversion (40% vs. 8%; *p* = 0.02). In Cohort 2, 166 patients were included in a symptoms-survey and tested for SARS-CoV-2. Any type of potential COVID-19-related symptom was referred up to 67.4% of patients (85.9% vs. 48.2% vs. 73.9%, for patients on chemotherapy, immunotherapy and targeted therapies respectively, *p* < 0.05). The seroprevalence ratio was 1.8% for the whole cohort with no significant differences by patient or treatment characteristics. Conclusion: Patients with cancer present higher risks for hospital needs for COVID-19 infection. The lack of SARS-CoV-2 seroconversion may be a concern for patients on immunosuppressive therapies. Patients receiving systematic therapies relayed a high rate of potentially COVID-19-related symptoms, particularly those receiving chemotherapy. However, the seroconversion rate remains low and in the range of general population.

## 1. Introduction

On 11 March 2020, the World Health Organization (WHO) declared a global pandemic for the Severe Acute Respiratory Syndrome Coronavirus 2 (SARS-CoV-2) Virus (coronavirus disease 2019, COVID-19) [1]. From that moment to the present time, hundreds of thousands of cases have been declared worldwide [2,3]. The report on the COVID-19 situation in Spain on 6 April 2020 revealed a total of 132,032 confirmed cases [4], although there are epidemiological studies such as the one published by Imperial College of the University of Oxford that estimates that on 30 March 2020, the infection rate in Spain could have been 15% (representing 7 million infected people, with a range between 1.8 and 19 million infected) [5].

Risk factors associated with worse prognosis of the infection are advanced age and comorbidities such as high blood pressure, cardiovascular diseases, respiratory diseases, diabetes or immune-suppression. All of them seem to increase the risk of hospitalization, need for admission to Intensive Care Units (ICUs), and death [6]. However, data on the risk of infection in cancer patients are very scarce. The first analysis of the risk of COVID-19 infection in cancer patients comes from the study by Liang et al, conducted on the Chinese population [7]. The authors analyzed a total of 1590 COVID-19 infected patients, of which 18 had a cancer history, representing 1% of the total, when the prevalence of cancer among the Chinese population is 0.29%. Furthermore, cancer patients had a higher risk of severe events compared to the rest of patients (39 vs. 8%, *p* = 0.0003), especially among those who had received chemotherapy treatment or had undergone recent surgical intervention. These findings led the authors to conclude that there is an increased risk of infection and poor prognosis for cancer patients, and consequently to recommend the suspension or postponement of chemotherapy treatments and surgical interventions in stable patients [7].

In Spain, we lack information regarding the association of COVID-19 and cancer. Despite the fact that this group of patients is considered frail and a population at risk, the diagnosis of cancer has not been included in the list of comorbidities that appear in official records [4]. On the other hand, it is not only important to know the incidence and severity of COVID-19 in cancer patients, but also the impact of different treatments, such as chemotherapy, immunotherapy, hormone therapy or targeted therapies, in the development and evolution of the disease. It is relevant to highlight that in the aforementioned article by Liang et al, no cases of cancer patients undergoing immunotherapy treatment are reported [7], despite the fact that immunotherapy represents a significant percentage of the treatments administered today in different types of cancers [8]. This fact may be due to chance, given that the sample of patients with cancer observed was very small, due to less use of immunotherapy in the Chinese population compared to the Spanish population, or perhaps due to a hypothetical protective effect of immunotherapy against the infection or the development of severe symptoms that motivated hospital admission for COVID-19. If this protective effect of the antibody anti-Programmed cell Death-1 receptor/Programmed Cell Death-1 Ligand 1 (anti-PD-1/PD-L1) inhibitors against COVID-19 infection is confirmed, it would open a way of investigation for the possible inclusion of these drugs in the therapeutic arsenal for the treatment of the disease.

The undetectable ‘iceberg’ of mild infections or asymptomatic, and non-immunized groups of patients needs to be estimated to fully assess disease severity. There are two principal types of diagnostic methods available: molecular and serological tests. At present, much attention is on the SARS-CoV-2 molecular test (Real Time polymerase chain reaction, RT-PCR). However, the RT-PCR test is not useful to differentiate between highly infective viruses versus ones that have been neutralized by the host, and it cannot estimate immunity status against SARS-CoV-2 [4]. Serological status established by antibody tests can complement RT-PCR tests and provide a more precise estimate of SARS-CoV-2 incidence and by potentially detecting subjects with immunity against the COVID-19 disease, as these tests detect biomarkers of the immune response. Such information is crucial to assess immunity status and is particularly helpful for appropriate decisions in cancer patients.

It is necessary to clarify the incidence of COVID-19 infection and its relationship with different cancer therapies to make decisions regarding patients’ treatment. For this reason, we have performed an observational study in order to define the incidence of COVID-19 in cancer patients in our environment, as well as to identify the possible impact that immune checkpoint inhibitor (ICI) immunotherapy might have on this population as a possible protector against infection.

## 2. Material and Methods

### 2.1. Study Design and Participants

The study has been carried out in a single institution, Hospital Arnau de Vilanova de Valencia of Spain. Two different cohorts of patients have been analyzed. Cohort 1. Patients with a medical history of cancer admitted to hospital with nasal and/or pharyngeal swabs positive test for SARS-CoV-2 RNA with Real Time polymerase chain reaction (RT-PCR) assay (Allplex™ Next generation of Real time PCR, Seegene Inc., Seoul, Korea) to perform a retrospective non-interventionist descriptive analysis of clinical and pathological characteristics and evolution of this population. Seroprevalence of Immunoglobulin M and Immunoglobulin G (IgM/IgG) antibodies were performed by ELISA-method (LIASON SARS-CoV-2 S1/S2 IgG test and LIASON SARS-CoV-2 IgM test, DiaSorin, Saluggia, Italy) and were carried out within two months from the second negative RT-PCR. Cohort 2. Patients with confirmed diagnosis of solid cancer treated at the outpatient medical oncology consultations with anticancer therapies between 2 May and 30 June 2020, to perform a non-interventionist cross-sectional study to assess seroprevalence for SARS-CoV-2 infection and assess history of symptoms related with COVID-19 and correlate them with demographic and pathological variables, and type of oncologic therapy administered.

### 2.2. Study Variables

Clinical variables related to COVID-19 infection collected were: symptoms (cough, fever, dyspnea, sore throat, chills, vomiting, diarrhea, constitutional syndrome, and anosmia), severity and duration of symptoms, need for hospital admission, need of intensive care, treatment received and outcome (cure or death). Analytical variables collected were: hematological cell counts, lactate dehydrogenase (LDH) levels, coagulation parameters, C-Reactive Protein at day 1 of hospital admission (cohort 1) or at the moment of inclusion in study (cohort 2). Serological analysis for COVID-19 infection was performed by Enzyme-Linked ImmunoSorbent Assay (ELISA) method (LIASON SARS-CoV-2 S1/S2 IgG test and LIASON SARS-CoV-2 IgM test, DiaSorin, Saluggia, Italy). Clinical retrospective data for patients included in cohort 1 were obtained from medical records, including demographic and clinical characteristics, and laboratory parameters. Patients included in cohort 2 were asked to complete a survey of symptoms presented during the previous two months, as well as their duration and severity, and the need of medical assistance or hospitalization.

### 2.3. Statistical Analysis

Descriptive statistics were obtained for two continuous variables (age and symptom duration) and 23 categorical variables. Descriptive tables and charts were obtained aiming at exploring the relationship between the type of disease treatment (chemotherapy, immunotherapy, or targeted therapy), the presence and severity of symptoms compatible with COVID-19, and the serology and RT-PCR results. Alongside this, appropriate hypothesis tests were performed, mainly for information purposes, given the reduced number of positive results for both the antibodies and the RT-PCR tests. More precisely, Fisher’s exact test [9] was used to assess the association between pairs of categorical variables, using Mehta and Patel’s [10] extension on cases other than 2 by 2 contingency tables when needed. Wilcoxon’s rank sum test [11] was used to quantify the differences in the distribution of a continuous variable (symptoms duration, in our case) between the two groups defined by a categorical variable. All analyses were performed with R 4.0.2 (R Core Team, Vienna, Austria) [12] on a computer running a Windows operating system.

### 2.4. Ethical Considerations

Ethics Committee of Hospital Arnau de Vilanova de Valencia approved this study: Ethics Committee No. HAV-BAR-2020-03. At all times, the current legislation on data confidentiality will be followed: Organic Law 03/2018 on Data Protection of 5 December 2018, published in BOE no. 294, BOE-A-2018-16673”.

The authors have no relevant affiliations with any organization or entity with a financial interest in the subject matter or materials discussed in the manuscript. All enrolled patients in the transversal study signed informed consent. The ethics committee approved the management of the retrospective study without obtaining the informed consent from patients who died or those whose evolution did not allow us to collect it.

## 3. Results

### 3.1. Retrospective Study of Patients with Cancer Hospitalized with PCR Confirmed Diagnose of COVID-19 (Cohort 1)

Between 28 February 2020 to 30 April 2020, 215 patients were admitted in our institutions with a RT-PCR-confirmed COVID-19 infection, and 17 of them died (7.9%). A total of 43 patients (20%) had a history of cancer. The median age was 73 years, 28 were male (65%), and the most frequent cancer types were hematologic (25.6%), prostate (21%), breast (9.3), bladder (9.3%) and colorectal (9.3%). There were 37 subjects found symptomatic (86%) and 6 were asymptomatic (14%) to COVID-19. There were 29 patients (67%) who were long-cancer survivors; 33 (76.7%) had not received any cancer-related treatment in the previous three months. Finally, ten subjects (23.2%) were receiving active treatment for cancer at the time of COVID-19 infection: 4 targeted therapy, 1 endocrine therapy, and 5 chemotherapies. Patients on active treatments for cancer represented 4.7% of all COVID-19-related admissions.

Six patients (14%) required admission to the Intensive Care Unit, and seven patients died from COVID-19 (16%); none of them were on systemic therapy for cancer (Table 1). Among the survivors, the RT-PCR clearance was confirmed at a median of 24+/− 15 days from initial diagnosis. Seroprevalence was determined in 36 surviving patients (Table 1). Thirty patients showed seroconversion (83%), 22 IgM negative/IgG positive (IgM−/IgG+) (61%), and 8 IgM positive/IgG positive (IgM+/IgG+) (22%), while 6 remained IgM negative/IgG negative (IgM−/IgG−) (17%). Four of the 6 patients with a lack of seroconversion were receiving systemic therapy for cancer: Cisplatin-based chemotherapy in 2 cases and Rituximab in 2 cases, and the other 2 had mild COVID-19 disease. The lack of seroconversion was significantly higher among cancer patients receiving systemic therapies than the rest of the cancer patients (40% vs. 8% respectively; *p* = 0.02).

### 3.2. Transversal Study of Patients Treated with Anticancer Therapies. Cohort 2

Between 2 May to 30 June 2020, 166 patients treated with ongoing anticancer therapy were included. Table 2 summarizes clinical and pathological characteristics. Median age was 62.3 years; 96 were male (57.8%). Seventy-eight patients received chemotherapy (46.9%), 58 were on exclusive immunotherapy (34.9%) and 23 on a targeted therapy (13.8%). The most frequent cancer types were lung (29.9%), breast (18,6%), colorectal (15,5%), and bladder (5.4%). Most patients presented with stage IV disease (n = 83.5%) or stage III disease (n = 13.4%).

A total of 112 active cancer patients (67.4%) reported a potential COVID-19-related symptom during the previous two months to study inclusion (Table 2). The most commonly reported were: asthenia 46.3%, cough 22.8%, headache 16.8%, dyspnea 16.8%, anosmia 16.2%, odynophagia 14.4%, fever 12.6%, and vomits 7.83% (Figure 1). Thirty-five patients (21.8%) reported basal activity reduction and 26 (15.6%) required medical attention. Ten patients (6%) required hospitalization in the two month period and 30 (18.7%) were tested for COVID-19 RT-PCR by medical criteria with one positive case (0.6%). Patients on chemotherapy treatment who referenced any COVID-19 potentially related symptoms were significantly higher than patients on immunotherapy and targeted therapies (85% vs. 48.2% vs. 3.9%, respectively, *p*-value < 0.05). These differences between chemotherapy, immunotherapy, and targeted therapies were due to a higher ratio of vomits (15% vs. 1.7% vs. 0.0%), diarrhea (32% vs. 6.9% vs. 21.7%), asthenia (58.9% vs. 36.2 vs. 43.4%), anosmia (24.3% vs. 5.1% vs. 21.7%) and headache (20.5% vs. 8.6% vs. 30.4%); there were no significant differences in cough, fever, dyspnea, or odynophagia (Figure 1).

Neither were differences in symptoms-duration, activity reduction, medical consultation or hospitalization according to type or anticancer therapy (Figure 2).

A total of 3 patients (1.8%) had positive seroconversion against SARS-CoV-2. Two were on chemotherapy treatment, and the other one on targeted therapy, which represents a seroconversion rate of 2.4% for the chemotherapy group, 4.1% for targeted therapies, and 0% for the immunotherapy group (differences not statistically significant). The only symptom significantly associated to positive seroconversion was fever (10% vs. 0.75%, *p*-value < 0.05). There were no statistically significant differences between patients with positive and negative seroconversion according to symptoms duration (mean 7 vs. 3 days).

## 4. Discussion

Recent series have pointed out the increased risk for COVID-19 infection and aggressiveness among cancer patients [7,13,14,15]. At first sight, our series is in line with those reports; a high number of admitted patients had a history of cancer (18.1%), with a higher need of Intensive Care (15.3%) and risk of death (18%). Our data coincide with those published recently by Venkatesulu et al. [16], as these authors performed a meta-analysis and systematically reviewed 31 studies that included 181,323 patients with COVID-19 disease, 23,736 of whom had history of cancer. They found that the subgroup of patients with cancer had an increased risk of death (16 vs. 5.4%, *p* = 0.0009), and a higher risk of developing Severe Acute Respiratory Syndrome (SARS) (OR 2.59, *p* = 0.004) when compared with the global population [16]. However, a deeper analysis of our cohort identifies that most cancer patients were elderly, long-term survivors, with other concurrent, mostly respiratory, comorbidity factors. In fact, among patients on systemic treatment for cancer, the prevalence was low and the severity generally mild. None of the nine patients on systemic therapy for cancer required intensive care or died from COVID-19-related disease.

Acquisition of immunogenicity against SARS-CoV-2 infection was observed in 83% of patients with cancer history hospitalized with confirmed infection by RT-PCR. It is remarkable that seroconversion was conditioned by the standard cytotoxic or immunosuppressive anticancer agents, since up to half of patients on active therapy failed to generate an adequate immunological response to COVID-19. Similar results have been reported on the immunogenicity of influenza vaccine for patients on Rituximab or chemotherapy [17,18]. This suggests that the immunosuppression induced by such therapies modulates the response to SARS-CoV-2 virus.

Another interesting finding is that we failed to identify any patient on specific immune-check point inhibitor (ICIs) admitted with COVID-19-related symptoms. At present, up to 30% of cancer patients in our institution are receiving an ICI regimen. Liang et al. showed that patients with cancer undergoing chemotherapy or surgery in the past month have risk of severe complications in comparison with subjects that are not receiving recent therapy. Curiously, none of the subjects received ICIs treatment [7]. In addition, some reports suggest that patients on ICI treatments might generate an immune-system reinforcement that would modulate response and symptoms to virus infections, such as SARS-CoV-2 virus [19,20]. Our translational study identified a 1.8% seroprevalence for SARS-Cov-2 antibodies in cancer patients receiving any anticancer therapy, which is similar to the 2% of seroprevalence in global population in the area of Valencia, recently published [21]. Despite the intensity of the pandemic suffered in Spain, only 5% of the population presents seroconversion, with differences by regions ranging from 10% in Madrid to 2% in seaside areas, such as ours. They also reported a higher ratio of seroconversion in patients who presented symptoms compatible with COVID-19 vs. asymptomatic (16.9% vs. 2%), and in RT-PCR-confirmed infection vs. non-tested (90% vs. 4%). These data suggest that seroconversion might depend on the intensity of the infection, and agrees with our finding of a lower rate of seroconversion in asymptomatic or mildly symptomatic patients vs. severe symptomatic patients with RT-PCR-confirmed infection.

Patients under anticancer treatment might present many of these symptoms as a side effect of the therapy, which makes the interpretation of our results difficult. Nevertheless, it is remarkable that during the pandemic period a significant number of patients presented COVID-19-related symptoms, especially in the subgroups of patients treated with chemotherapy and targeted therapies. The ultimate cause of the low rate of seroconversion in our patients remains unclear. It might be due to a greater protection in isolation and compliance with social distances measures in this special population. However, it could also be produced by the deleterious effect of chemotherapy and other immunosuppressive therapies used for cancer treatment, which as we have found in the retrospective cohort could impact a deficient immune response to the viral infection. In addition, previous studies have shown that circulating antibodies against SARS-CoV or MERS-CoV last for one year [22,23]. IgG levels were sustained for more than two years after SARS-CoV infection [24]. However, one mathematical model suggests a short duration of immunity after SARS-CoV-2 infection [25].

## 5. Conclusions

Overall, we have not found a greater risk of COVID-19 infection in patients with cancer under oncological treatment when compared with the global population, nor a worse outcome for those patients hospitalized with COVID-19 disease. However, the immunosuppressive effect of some anticancer therapies could impact the immune-system response efficacy against the virus. International guidelines are recommending molecular and serological screening for SARS-CoV-2 infection in patients undergoing systemic cancer therapy. Our findings stress the importance of close monitoring of patients recovered from COVID-19 and particularly in the absence of serologic conversion. For this reason, our intention is to monitor over time the developed immunity in order to determine its durability and effectiveness.

## Figures and Tables

**Figure 1 cancers-12-03513-f001:**
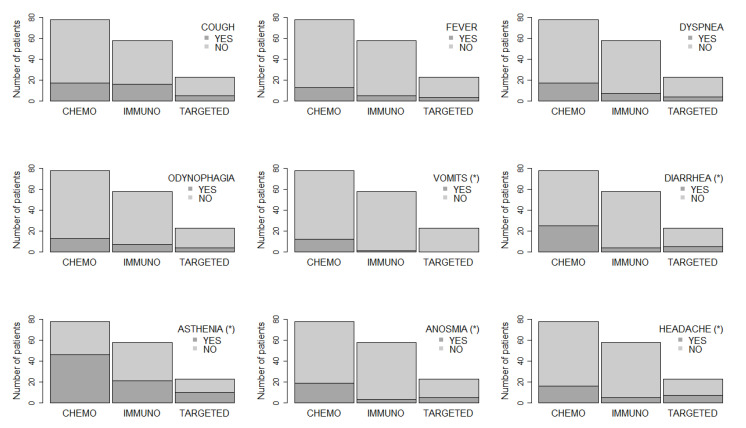
Relationship between the type of treatment and the presence or absence of symptoms. A star (*) means that there is a statistically significant association according to Fisher’s exact test [9] (sig. level = 0.05).

**Figure 2 cancers-12-03513-f002:**
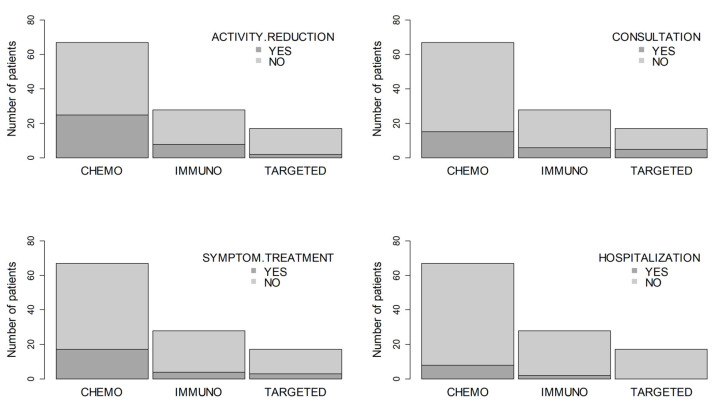
Relationship between the type of treatment and the severity of symptoms. No statistically significant associations were found according to Fisher’s exact test [9] (sig. level = 0.05).

**Table 1 cancers-12-03513-t001:** Demographic, clinical characteristic, and seroprevalence of COVID-19 infected cancer patients. Cohort 1.

Characteristic	Overall Cohort with COVID-19 Number (%) (n = 43)
Age (median)	73 (46–91)
Male	28 (65)
**Clinical Expression**	
Symptomatic	37 (86)
Asymptomatic	6 (14)
**Cancer type**	
Breast	4 (9.3)
lung	2 (4.7)
Prostate	9 (21)
Kidney	3 (7)
Bladder	4 (9.3)
hematology	11 (25.6)
Colon	4 (9.3)
other	6 (14)
**Outcomes**	
Hospitalization	43 (100)
Severe events	14 (33)
Intensive care unit	6 (14)
Death	7 (16)
**Cancer Status**	
Remission or no evidence of disease	29 (67)
Present, stable	11 (25.6)
Present and metastasis	3 (7.7)
**Anticancer Therapy**	
None ^a^	33 (76.7)
Target therapy ^b^	4 (9.3)
Hormonotherapy	1 (2.3)
Immunotherapy	0
Chemotherapy ^c^	5 (12)
Radiotherapy	**0**
**Seroprevalence** ^**d**^	n = 36
IgM−/IgG+	22 (61)
IgM+/IgG+	8 (22)
IgM−/IgG−	6 (17)

Data were summarized as percentages and mean with standard deviation or as median with interquartile range (IQR). Data analysis was performed on 10 June 2020. ^a^ No systemic therapy within the 3 months before COVID-19 diagnosis. ^b^ Immunosupressants (Rituximab, Ustekinumab), ^c^ CDDP-Gemcitabine, Folfoxiri, CAPOX-Bevacizumab. ^d^ Serological study was not performed in the seven dead patients.

**Table 2 cancers-12-03513-t002:** Demographic, clinical characteristic and seroprevalence of COVID-19 in patients receiving anticancer treatment. Cohort 2.

Characteristic	Overall Cohort Patients Number(%) (n = 166)
Age (median)	63 (33–86)
Male	96 (57.8)
**Symptoms**	
Symptomatic	112 (67.4)
Asymptomatic	7 (32.6)
**Cancer type**	
Breast	31 (18.6)
lung	49 (29.5)
Colon	20 (12.5)
Bladder	9 (5.4)
Gastric	6 (3.61)
Rectum	6 (3.6)
Ovary	7 (4.2)
ORL Hepatocarcinoma Pancreas Other	6 (3.6) 4 (2.4) 5 (3.0) 23 (13.8)
**Stage**	
Stage IV	137 (83.5)
Stage III	22 (13.2)
Stage I-II	5 (3.0)
**Anticancer Therapy**	
Chemotherapy	78 (46.9)
Immunotherapy	58 (34.9)
Targeted Therapy	23 (13.8)
**Symptoms**	
Cough	38 (22.8)
Feber	21 (12.6)
Dyspnea	28 (16.8)
Odynophagia	24 (14.4)
Vomits	13 (7.8)
Diarrhea Asthenia Anosmia Headache Activity Reduction Medical Consultation Treatment Required Hospitalization	34 (20.4) 77 (46.3) 27 (16.2) 28 (16.8) 35 (21.0) 26 (15.6) 24 (14.4) 10 (6.2)
**Seroprevalence**	
IgM−/IgG+	2 (1.2)
IgM+/IgG+	1 (0.6)
IgM−/IgG−	163 (98.1)

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
