# Peer review of "Impact of SARS-CoV-2 Infection on Patients with Cancer: Retrospective and Transversal Studies in Spanish Population"

_cancers, 2020, doi:10.3390/cancers12123513_

Round 1
Reviewer 1 Report
Authors describe SARS-CoV-2 infection and seroconversion in patients with cancer in Spain. The question is of interest. An ethics vote is present. The patient group is sufficiently large. Study variables fit clinical setting. 43 of 215 CoV-2 patients suffered from cancer. Based on investigated patients' number, most relations between treatment or side effects were not significant. Symptoms of cancer and cancer therapy make COVID diagnostics more challenging. In summary, it is a sound and well presented clinical analysis.
Reviewer 2 Report
The authors presented interesting data on the correlation between the incidence and severity of SARS-Cov-2 infection in patients with cancers. Additionally they analyzed the influence of anticancer therapies on the immune-system response to the virus. Although this is a one-centre observation, the results improve our knowledge in this currently very important field. I have some minor remarks:
- Please, provide the methods use for lab tests.
- Please, provide the information about patients consents to the participation in the study and publication of the results
- Please, correct several linguistic and typing errors with the assistance of a native speaker
- Please, re-preapare the manuscript according to the Instruction for Authors (reference style, abstract lenght, etc.)
